# Impact of National Tobacco Control Policy on Rates of Hospital Admission for Pneumonia: When Compliance Matters

**DOI:** 10.3390/ijerph20105893

**Published:** 2023-05-20

**Authors:** Marine Gambaryan, Anna Kontsevaya, Oxana Drapkina

**Affiliations:** National Medical Research Centre for Therapy and Preventive Medicine of the Ministry of Health of Russia, 101990 Moscow, Russia; akontsevaya@gnicpm.ru (A.K.); odrapkina@gnicpm.ru (O.D.)

**Keywords:** tobacco control, tobacco control legislation, tobacco control measures, tobacco control implementation scale, interrupted time series design, hospitalisation rates for pneumonia, smoking-related hospitalisation rates

## Abstract

A number of studies claim that tobacco control (TC) regulations are associated with reductions in smoking-related hospitalisation rates, but very few have estimated the impact of TC laws (TCL) at both countrywide and regional levels, and none of them have studied the impact of TCL in relation to compliance with TC regulations. This study evaluates the effects of Russian TCL on hospital admission (HA) rates for pneumonia countrywide and in 10 Russian regions and the extent of these effects in connection with the compliance with TCL. Methods: HA rates for pneumonia from 2005–2019 were analysed to compare the periods before and after the adoption of TCL in 2013. An interrupted time series design and a Poisson regression model were used to estimate the immediate and long-term effects of TCL on pneumonia annual hospitalisation rates after the TCL adoption, compared with the pre-law period. The 10 Russian regions were compared using the TCL implementation scale (TCIS) developed on the basis of the results of the Russian TC policy evaluation survey; Spearman’s rank correlation and linear regression models were employed. Results showed a 14.3% reduction in HA rates for pneumonia (RR 0.88; *p* = 0.01) after the adoption of TCL in Russia with significant long-term effect after 2013 (RR 0.86; *p* = 0.006). Regions with better enforcement of TCL exhibited greater reductions in pneumonia HA rates (rsp = −0.55; *p* = 0.04); (β = −4.21; *p* = 0.02). Conclusions: TCL resulted in a sustained reduction in pneumonia hospitalisation rates, but these effects, varying by region, may depend on the scale of the TCL enforcement.

## 1. Introduction

Smoking is known as a risk factor for many cardiovascular and respiratory diseases: active and passive smoking increase morbidity and premature mortality from lung cancer, acute coronary syndrome and respiratory diseases [1,2]. Smoking-related diseases are a cause of premature mortality and account for 87% of the total mortality in Russia [3].

A growing body of evidence from different countries indicates that tobacco control (TC) regulations, including bans on smoking in public places, can reduce hospital admissions for acute cardiovascular disease [4,5,6,7] and respiratory diseases [8,9,10], along with affecting the incidence of lung cancer in the long term [11,12].

Several studies have discovered a reduction in hospital admission (HA) rate for acute lower respiratory tract infections in children, following the introduction of smoking bans in public places and other TC regulations [13,14,15,16]. Evidence of the effect of TC provisions on respiratory diseases in the general population showed mainly a reduction in hospital admission rates for exacerbations of chronic bronchitis, COPD or asthma [17], while very few studies did so in relation to the admissions for pneumonia in adults [18].

Existing studies have primarily assessed the impact of smoking bans in indoor public places on hospital admission rates. However, other TC measures may also affect hospital admission rate for smoking-related acute conditions. Moreover, all of these studies evaluated the direct impact of TC measures on the monthly rate of hospital admissions for smoking-related conditions [5,13,14,15] over one or two years after the introduction of such laws, making it difficult to verify the long-term effects of TC legislation. Finally, very few studies have estimated the impact of national TC legislation at both countrywide and regional levels [5,13,14,15], and none of them have done so in relation to compliance with these regulations.

In 2013, Russia introduced one of the most comprehensive TC laws in Europe, aimed at protecting public health from the effects of tobacco smoke and the consequences of smoking. In accordance with the provisions of the WHO Framework Convention on Tobacco Control (FCTC), the Russian Tobacco Control Law (RTCL) introduced a complete ban on smoking in indoor and outdoor public places, workplaces and all public transport; a ban on advertising, promotion and sponsorship of tobacco products, including open display of tobacco products at points of sale; restrictions on the retail sale of tobacco products, e.g., near educational facilities; annual increase in excise taxes on tobacco products; information-communication measures to raise awareness of health risks associated with tobacco use and passive smoking; and measures to help smokers quit smoking. These regulations aim to reduce the prevalence of smoking as well as smoking-related morbidity and mortality in the population. Despite the notable progress made by Russia after the ratification of the FCTC, the prevalence of smoking is still high, at 27.3%: 46.4% in men and 14.6% in women in 2018 [19,20]. The slowdown in the rate of decrease in smoking prevalence by only 3.5% compared with 2013, i.e., after five years of RTCL implementation, should be noted [19,20]. However, both the prevalence of smoking and its changes varied across different regions of Russia [19,20].

Hence, we hypothesized that the effects of RTCL on the prevalence of smoking and its impact on public health may depend on the degree of enforcement and implementation of these regulations in the regions of Russia. We evaluated the implementation of the law by measuring the compliance with the TC regulations. With this goal in mind, we conducted an evaluation survey in 2017–2018 based on a random sample of 11,625 participants in 10 constituent entities of the Russian Federation. Our Russian Tobacco Control Policy evaluation survey was representative both of the entire country and of the regions. This survey established different levels of compliance with TC measures across the 10 regions. This survey established different levels of compliance with TC measures across the 10 regions. In some of these regions, e.g., Chuvash Republic, the results of the survey showed high compliance with smoking bans, existence of tax and price policies, and comprehensive support of smoking cessation. In other regions, such as Arkhangelsk Oblast, these measures were implemented to a lesser extent. Using the scoring system of the Tobacco Control Scale by L. Joossens and M. Raw [21] and the results of our Russian Tobacco Control Policy evaluation survey, we developed an original scale to assess the implementation of the most cost-effective existing TC measures (viz., the WHO MPOWER provisions) in the regions of Russia, and we have named it the Tobacco Control Implementation Scale (TCIS) [22]. Our previous studies demonstrated a significant relationship between TCIS scores and changes in smoking prevalence in 10 regions over a five-year period after the introduction of RTCL [19].

Our two previous studies assessed changes in monthly hospitalization rates for acute coronary events (ACE) in three regions of Russia and annual hospital admission rates for ACE in the entire country and 10 regions of Russia [23,24]. The latter study [24] showed a significant reduction in hospital admissions for both angina and myocardial infarction (16.6% [RR 0.83, 95% CI 0.74–0.93] and 3.5% [RR 0.96, 95% CI 0.96–0.97], respectively) after the nationwide introduction of RTCL vs. the period before the adoption of this law, as well as effects of varying magnitude in 10 regions. Regions with better enforcement of the TC law experienced greater reductions in hospital admission rates for angina and myocardial infarction.

In this study, we sought to analyse the impact of Russian TC regulations on hospital admission rates for all-cause pneumonia in adults, which may be especially important in an era of the COVID-19 pandemic or other viral infections. In addition, we intended to analyse the extent of these effects in various regions of Russia in connection with the implementation of TC legislative measures.

## 2. Materials and Methods

### 2.1. Study Design and Data Sources

We analysed hospitalization rates for all-cause pneumonia to compare periods before and after the adoption of RTCL in 2013, adjusting for possible confounding factors and long-term trends. We used an interrupted time series analysis to quantify the change in hospital admissions for pneumonia after the adoption of RTCL vs. the preceding period. To demonstrate that immediate and gradual changes in hospitalisation rates for pneumonia were associated with RTCL, we also analysed asthma hospitalization rates for comparison. We assumed that since asthma has been routinely and legitimately monitored and controlled for many years at the outpatient level, the effect of RTCL on hospital admission rates for asthma should not be apparent. We also analysed hospitalization rates for rheumatic heart disease (RHD) for comparison (as a disease not associated with smoking), which should not be affected by TCL, and which has also been routinely controlled like asthma at outpatient level. The models were based on the time series of annual hospitalizations for all three diseases in the Russian Federation and 10 regions over the period of 2005–2019.

We also analysed the change in hospital admission rates after the adoption of RTCL, compared with the pre-law period in 10 regions of Russia, depending on the degree of enforcement of the TC regulations. To compare regions with different levels of compliance with RTCL, we used the TCIS developed in our previous study [22].

Annual data on hospital admissions for pneumonia and bronchial asthma in the adult population that occurred in Russia and its 10 regions between 2005 and 2019 were obtained from the national official hospital discharge statistics database, which included the following information: the diagnosis at discharge, age category (0–17, including 0–1, and ≥18 years old), and region of residence. Respiratory outcomes were defined according to the International Classification of Diseases—10 codes for all-cause pneumonia (J12–18) and asthma (J45, J46).

We analysed the changes in the rate of hospitalization for pneumonia per 100,000 adult residents (aged ≥18 years) in the Russian Federation and its 10 constituent entities: the Chuvash Republic, Krasnodar and Primorsky Krais, and Arkhangelsk, Astrakhan, Belgorod, Novosibirsk, Orenburg, Samara and Tyumen Oblasts.

Smoking prevalence in the years 2013, 2018 and 2019 was taken from the population surveys database of the Federal State Statistics Service; population data by age group (0–17 years and ≥18 years), as well as data on as hospital bed-population ratio, were taken from the official statistics of Federal State Statistics Service [20,25,26].

### 2.2. Tobacco Control Implementation Scale

To compare changes in the hospital admission rate for pneumonia before and after the adoption of RTCL in different regions in terms of adherence to TC regulations, we used the scores of TCIS [22]. The scale indicates how well the six MPOWER activities were implemented in each investigated region. Table A1 in Appendix A shows how the scale applies to the 10 Russian regions, where the investigation was carried out and the ranks of the regions according to the scores of TCIS, characterising the performance of the MPOWER measures in each of the region.

The scores characterising the performance of the MPOWER package and each of its six measures were used as independent variables in the correlation and linear regression analyses [19,22].

### 2.3. Statistical Data Processing

We used standard methods for interrupted time series (ITS) to evaluate the effects of RTCL [27]. The immediate effect was modelled as a step function including an indicator variable that changed after 2013, whereas the gradual effect was investigated via an interaction term between the RTCL impact and time. We employed a generalized Poisson regression model with calculation of the incidence rate ratio (RR) and a 95% confidence interval (95% CI) to estimate the immediate and long-term effects of RTCL.

The following regression model was used:Yt = β_0_ + β_1_T + β_2_Xt + β_3_TX
where Yt represents the outcome at time t; T is the time elapsed since the start of the study in with the unit representing the frequency with which observations are taken (year); Xt is a dummy variable indicating the pre-intervention period (coded 0) or the post-intervention period (coded 1); and β_0_ represents the baseline level at T = 0, β_1_ is interpreted as the change in outcome associated with a time unit increase (representing the underlying pre-intervention trend), β_2_ is the level change following the intervention and β_3_ indicates the slope change following the intervention (using the interaction between time and intervention: TXt).

Scaling corrections were applied to the model to avoid overdispersion and misestimation of standard errors. The models were also tested for autocorrelation.

To assess the relationship between the relative change in hospital admission rates after the adoption of RTCL (%), smoking prevalence rates in each region (dependent variable) and scores characterizing the degree of implementation of TC legislative measures (independent variable), we performed Spearman’s rank correlation analysis and linear regression analysis.

The analyses were carried out using the Stata v.11.2. statistical software (StataCorp, Lakeway, TX, USA).

## 3. Results

A total of 5,785,673 hospital admissions for pneumonia and 2,575,561 for bronchial asthma occurred among the Russian population during the study period. Of these, 2,395,953 cases of pneumonia and 865,994 cases of asthma were detected after the adoption of RTCL in 2013.

Figure 1 presents data on annual age-adjusted hospital admission rates for pneumonia, bronchial asthma and rheumatic heart disease per 100,000 residents from 2005 to 2019 in Russia. These data are shown in Figure 1 by year, along with the predicted regression curves. The figure demonstrates the trends in annual hospital admission rates for pneumonia (1A) and asthma (1B) in the Russian Federation from 2005 to 2019, i.e., before and after the introduction of RTCL in 2013. Dynamics of hospitalization rates after the introduction of RTCL (solid line) in comparison with the predicted trend without TC measures (dashed line) is shown in Figure 1 as well.

We observed a significant decrease in hospital admission rates for pneumonia by 14.3% after the adoption of RTCL (2014–2019), compared with the pre-law period (2005–2013): the RR was 0.88 (95% CI 0.79–1.00) (*p* = 0.01).

We also revealed evidence of a gradual effect of RTCL; change in the main trend of hospital admission rate for pneumonia after 2013: RR = 0.86 (95% CI 0.77–0.96) (*p* = 0.006).

As for asthma and rheumatic heart disease, there was no statistically significant reduction in hospital admission rates after the adoption of RTCL vs. the preceding period: RR = 1 (95% CI 0.97–1.1) (*p* = 0.779) and RR = 0.94 (95% CI 0.83–1.06) (*p* = 0.332).

Secondary analyses conducted among the adult population in 10 constituent entities of Russia yielded similar effects of RTCL in different regions. However, these effects had different magnitudes, and the decrease in the hospital admission rates for pneumonia after the adoption of RTCL was statistically significant in only 4 out of 10 regions (Table 1).

Moreover, adjusting for factors potentially affecting hospital admission rates, such as hospital bed-population ratio, did not significantly change the results.

Table 1 demonstrates changes in hospital admission rates for pneumonia after the adoption of RTCL, compared with the preceding period, in 10 regions of the Russian Federation.

Thus, we hypothesized that the degree of reduction in hospital admission rates after the adoption of the RTCL vs. the preceding period in the regions may be related to the degree of this law enforcement.

To check this hypothesis, we measured the correlations and the associations between the relative changes of hospital admission rates for all three conditions and the TCIS scores characterising the extent of the implementation of TCL measures in the 10 regions by conducting Spearman’s correlation analysis and a linear regression analysis.

Table 2 presents the correlations of reduction in the rate of hospital admissions for pneumonia (RR%) with the degree of implementation of six MPOWER measures in 10 regions of Russia based on the TCIS scoring system (Appendix A, Table A1).

We detected significant correlations between the reduction in hospital admission rates for pneumonia and TCIS scores for smoking ban rsp = −0.55 (95% CI −1.08, 0.02) (*p* = 0.042) and for offering support in smoking cessation: rsp = −0.763 (95% CI −1.11, −0.41) (*p* < 0.001).

An inverse correlation was also established between the decrease in hospital admission rate for pneumonia and the prevalence of smoking in 2019 in the regions: rsp = 0.7 (95% CI −0.08, 2.25 (*p* < 0.05).

Linear regression analysis yielded significant associations between the decrease in hospital admission rates for pneumonia (RR%) and the TCIS score for offering smoking cessation support (β = −4.21; 95% CI −7.61, −0.82; *p* = 0.02), as well as with the prevalence of smoking in 2019 (β = 2.40; 95% CI 0.34, 4.45; *p* = 0.027). Both relationships were significant for pneumonia, but not for asthma (Table 3).

We did not reveal any statistically significant correlations or significant associations between the reduction in the rates of hospital admissions for asthma and rheumatic heart disease and the extent of implementation of either TCL measures or smoking prevalence in 10 regions of Russia.

## 4. Discussion

Our analysis, based on nearly six million hospital admissions, showed for the first time the long-term (over 15 years) trends of annual hospitalisation rates for all-cause pneumonia. We established a reduction in the rates of hospital admission for all-cause pneumonia among the adult population over the entire study period after the adoption of a comprehensive TC law in Russia. The observed decrease was similar across regions of Russia and was stronger in constituent entities with better compliance with TC regulations.

Our findings are consistent with several previously published studies demonstrating a reduction in hospital admissions for acute lower respiratory tract infections associated with adoption of a municipal or national smoking ban. A. Nyman et al. observed a 33% decrease in hospital admissions for respiratory diseases during a restaurant ban in Toronto [17]. A study by J.-P. Humair et al. demonstrated that smoking bans resulted in a very significant reduction in hospitalizations for exacerbations of COPD and no significant changes in hospital admissions for pneumonia and acute asthma in the Canton of Geneva [9]. However, changes in hospitalization rates in these studies were limited to the short period of smoking ban introduction and did not extend to the longer period after the ban. V. Ho et al. described the association between smoking bans, as well as higher excise taxes on cigarettes, with reduced rates of hospital admission for pneumonia in individuals 60 to 74 years of age in a nationwide study conducted in the USA [18].

Unlike most previous studies on smoking bans, we measured the relationship between implementation of a comprehensive TC law and hospitalization rates nationwide. In addition, by analysing data on the implementation of RTCL in 10 constituent entities of the Russian Federation, we determined which of the TC measures had the greatest impact on reducing hospitalization rates for pneumonia among adults.

In our study, we assessed the gradual effect of reduced hospital admissions for pneumonia depending on the degree of enforcement of RTCL in 2017–2018. Our results implied that more effective implementation of anti-tobacco measures in the regions and the degree of their enforcement in 2017–2018 (assuming they were similar from the first year of RTCL adoption) could affect the change in rates of hospitalization for pneumonia.

Because we were looking at annual rather than monthly hospitalization rates, there was no need to adjust the models for seasonality. However, we adjusted the model for potential confounders, such as hospital bed-population ratio, which did not affect the results in any way.

TC regulations aim to reduce the prevalence of smoking and smoking-related morbidity and mortality in the long term. L. Palmieri et al. demonstrated a reduction in smoking prevalence in Italy from 31.7% to 21.8% between 1980 and 2000, which led to a decrease in mortality from coronary heart disease [28]. Smoking is a risk factor for developing pneumonia. A meta-analysis by V. Baskaran et al. showed that current smokers and ex-smokers were 2.7 and 1.5 times, respectively, more likely to develop community-acquired pneumonia, compared to “never smokers” [29]. Their other finding was that current heavy smokers had a significantly higher risk of developing pneumonia than light smokers.

Our study suggests that greater relative changes in smoking prevalence over the five-year period of RTCL implementation (2013–2018) and lower smoking prevalence in 2019 may be associated with lower hospitalization rates for pneumonia and better RTCL enforcement.

There are some strengths and limitations of the study that should be mentioned.

Among the strengths of this study, we should mention its large sample size, encompassing all nationwide hospital admissions for pneumonia over a 15-year period. It explored the immediate and long-term impact of comprehensive tobacco control legislation on all-cause pneumonia in adults, which were not previously studied. In addition, relationships between the impact of comprehensive RTCL and individual legislative measures on hospitalization rates and the degree of implementation of these measures in different regions, based on large representative survey data, were investigated, which increased the strength of our study as well.

The limited number of investigated regions can be seen as a limitation of the study in terms of its ability to explore possible relationships in correlation and linear regression analyses. However, despite the limited number of regions, we still revealed statistically significant relationships.

Another limitation of our study is related to the data of a representative survey of the population assessing the Russian Tobacco Policy in 10 constituent entities of the Russian Federation. We assumed that the degree of compliance with anti-tobacco regulations, measured in 2017–2018 in 10 regions of Russia, was similar over the entire study period starting from the first year of RTCL adoption.

## 5. Conclusions

The results of this study conducted on a large population sample over a long follow-up period suggest that a comprehensive tobacco control policy can lead to an immediate reduction in hospital admission rate for all-cause pneumonia with a gradual effect. This finding has important public health implications, especially in the era of the COVID-19 pandemic and/or other viral infections. Smoking regulations represent a simple, effective and inexpensive way to prevent respiratory diseases, and the degree of compliance with the regulations can be important for the prevention of these ailments.

## Figures and Tables

**Figure 1 ijerph-20-05893-f001:**
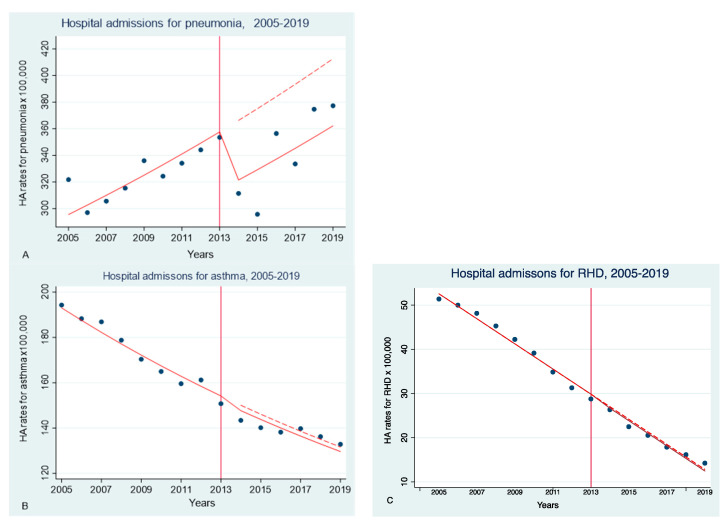
HA rates for pneumonia, asthma and RHD per 100,000 residents in the Russian Federation during the period of 2005–2019. Observed (solid lines) and predicted (dashed lines) adjusted HA rates for pneumonia (**A**), asthma (**B**) and RHD (**C**) in the adult population.

**Table 1 ijerph-20-05893-t001:** Changes in hospitalisation rates for all-cause pneumonia after the adoption of RTCL compared with the preceding period.

Federal Subjects of Russia	IRR * (95% CI)	*p*	Relative Change IRR (%)
The Russian Federation	0.88 (0.79–0.97)	0.011	−14.3
The Chuvash Republic	0.75 (0.62–0.91)	0.003	−27.5
Krasnodar Krai	0.98 (0.8–1.2)	0.863	−4.0
Primorskyi Krai	1.01 (0.99–1.03)	0.694	1.1
Arkhangelsk Oblast	0.88 (0.77–0.99)	0.04	−14.7
Astrakhan Oblast	0.88 (0.72–1.08)	0.224	−17.8
Belgorod Oblast	0.77 (0.6–0.98)	0.033	−27.4
Novosibirsk Oblast	0.96 (0.79–1.16)	0.681	−4.6
Orenburg Oblast	0.93 (0.76–1.14)	0.512	−10.4
Samara Oblast	0.72 (0.63–0.82)	0.000	−32.4
Tyumen Oblast	0.88 (0.67–1.2)	0.353	−13.2

* IRR—Incidence Rate Ratio

**Table 2 ijerph-20-05893-t002:** Correlation between changes in hospital admission rates for pneumonia (RR%) after the adoption of RTCL vs. the pre-law period, prevalence of smoking in adult population, and TCIS scores.

RTCL	rsp * (95% CI)	*p*
All MPOWER measures	−0.02 (−0.69; −0.66)	0.958
Tax/price measures	−0.11 (−0.62;0.86)	0.764
**Smoking bans ****	**0.55 (−1.08; 0.02)**	**0.042**
Information and communication measures	−0.127 (−0.84; 0.58)	0.725
Banning tobacco advertising, promotion, sponsorship	−0.004 (−0.745; 0.75)	0.990
Warning signs	0.40 (−0.22; 1.03)	0.208
**Smoking cessation support**	**−0.763 (−1.11; −0.41)**	**0.000**
**Prevalence of smoking**		
**Smoking prevalence in 2019**	**0.7 (0.32; 1.08)**	**0.000**
Changes in smoking prevalence 2013–2018	−0.5 (−1.07; −0.07)	0.085

* rsp—Spearman’s rank correlation coefficient; ** bold font designates statistically significant results: *p* < 0.05.

**Table 3 ijerph-20-05893-t003:** Association between changes in hospital admission rates for pneumonia and asthma (RR%) after the adoption of RTCL vs. the preceding period, and TCIS scores for smoking cessation support and smoking prevalence in 2019, identified by linear regression.

Reduction in Hospital Admission Rates (RR%)	Smoking Cessation Support	Prevalence of Smoking in 2019
	β * 95% CI	β 95% CI
Pneumonia	**−4.212 (−7.61; −0.82)** ******	**2.40 (0.34; 4.45)**
*p*	0.020	0.027
Asthma	−2.43 (−5.39; 0.53)	1.2 (0.64; 3.04)
*p*	0.096	0.174

* β—regression coefficient; ** bold font designates statistically significant results: *p* < 0.05.

## Data Availability

Data on annual hospital admissions are not public; they are obtainable from the official national official hospital discharge statistics database at the Data centre the Russian Research Institute of Health. For this study, hospital admission data were provided by the Russian Research Institute of Health upon official request. Data on smoking prevalence for the years 2013, 2018 and 2019 were obtained from Population Surveys of the Federal State Statistics Service, provided upon request. Population data by age group (0–17 years and ≥18 years) and data on hospital bed-population ratio were taken from the Federal State Statistics Service official database available on the website of the Federal State Statistics Service: https://gks.ru/bgd/regl/b20_14p/Main.htm. (accessed on 3 April 2023).

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
