# Peer review of "Impact of National Tobacco Control Policy on Rates of Hospital Admission for Pneumonia: When Compliance Matters"

_ijerph, 2023, doi:10.3390/ijerph20105893_

Round 1
Reviewer 1 Report
1. I agree the sample size is impressive, but all-cause pneumonia admissions are a broad group. Relating decrease in overall hospital admissions just to decreased smoking is hard, no matter how many statistical models we apply. That being said, I think methods adopted by the authors are valid and this sort of literature makes it easier to take smoking cessation efforts to policy makers.
2. Line 64, authors state level of implementation of smoking ban can be different in different regions, but it is not explained fully which regions (of the 10 regions picked) were less or more strict with the ban.
Acceptable
Author Response
Response to Reviewer 1 Comments
Report on IJERPH-2353846
Point 1: 1. I agree the sample size is impressive, but all-cause pneumonia admissions are a broad group. Relating decrease in overall hospital admissions just to decreased smoking is hard, no matter how many statistical models we apply. That being said, I think methods adopted by the authors are valid and this sort of literature makes it easier to take smoking cessation efforts to policy makers.
Response 1:
We appreciate your comment, thank you very much!
Point 2: Line 64, authors state level of implementation of smoking ban can be different in different regions, but it is not explained fully which regions (of the 10 regions picked) were less or more strict with the ban.
Response 2:
We assessed the implementation of the law by examining compliance as measured by the survey. Which regions out of 10 were less or more strict with the law, we could judge solely by the results of the survey. However, we have added a paragraph (lines 78-82) to highlight differences in law enforcement across regions.

Reviewer 2 Report
Report on IJERPH-2353846
The Russian government introduced comprehensive tobacco control laws (TCL) including smoking bans, high taxes, smoking cessation support and other measures in 2013. In this paper, the authors use the Russian tobacco control policy evaluation survey and administrative data and an interrupted time series model to estimate the impacts of TCL compliance on hospital admissions because of pneumonia. They found that adopting the TCL reduced the hospital admission rate of pneumonia patients by 14.3%. They also investigated the correlation between pneumonia admission and the compliance level of different tobacco control policies as pathways of the effect and document a significant negative correlation between the availability of smoking cessation support and the hospital admission rate and a significant positive correlation between smoking bans and the hospital admission rate.
This is a very important topic given the high smoking prevalence in Russia. I really enjoyed reading this paper. I have some comments below.
Background
Readers will see more clearly the importance of the research question if they understand better the past and current situation of smoking in Russia. I suggest that the authors add more background information about smoking rates and tobacco-related diseases in the country.
Method
Regarding the interrupted time series model, I think it is better to use the year fixed effects instead of the current T variable (the time elapsed since the start of the study) to avoid the assumption that the effect is linear over time. The authors should consider adding the region fixed-effects in the model since the effects across regions can vary substantially in a large country like Russia.
The author compared the hospitalization rates of pneumonia and asthma. First, asthma can be impacted by tobacco control laws as well. So I suggest that the authors use a non-tobacco-related disease for comparison. Second, currently the authors compared hospital admissions for pneumonia and asthma using graphs only (Figure 1 specifically). They can make a better comparison by running the same interrupted time series model for asthma or a non-tobacco-related disease, which I recommend, and see if the TCL has had any impacts on the control.
Results
The findings about the correlation between hospital admissions for pneumonia and different components of the TCL provide nice support to the conclusion. However, I recommend that the authors separate the main findings and the mechanism findings since the models used for the estimation are different, and also the channels of the effect should be highlighted more in the paper.
Author Response
Response to Reviewer 2 Comments
Report on IJERPH-2353846
The Russian government introduced comprehensive tobacco control laws (TCL) including smoking bans, high taxes, smoking cessation support and other measures in 2013. In this paper, the authors use the Russian tobacco control policy evaluation survey and administrative data and an interrupted time series model to estimate the impacts of TCL compliance on hospital admissions because of pneumonia. They found that adopting the TCL reduced the hospital admission rate of pneumonia patients by 14.3%. They also investigated the correlation between pneumonia admission and the compliance level of different tobacco control policies as pathways of the effect and document a significant negative correlation between the availability of smoking cessation support and the hospital admission rate and a significant positive correlation between smoking bans and the hospital admission rate.
This is a very important topic given the high smoking prevalence in Russia. I really enjoyed reading this paper. I have some comments below.
Point 1: Background
Readers will see more clearly the importance of the research question if they understand better the past and current situation of smoking in Russia. I suggest that the authors add more background information about smoking rates and tobacco-related diseases in the country.
Response 1: Background
More background information about smoking rates and tobacco-related diseases in Russia is added as suggested by the reviewer (lines 32-33; 64-75).
Point 2: Method
- Regarding the interrupted time series model, I think it is better to use the year fixed effects instead of the current T variable (the time elapsed since the start of the study) to avoid the assumption that the effect is linear over time. The authors should consider adding the region fixed-effects in the model since the effects across regions can vary substantially in a large country like Russia.
- The author compared the hospitalization rates of pneumonia and asthma. First, asthma can be impacted by tobacco control laws as well. So I suggest that the authors use a non-tobacco-related disease for comparison. Second, currently the authors compared hospital admissions for pneumonia and asthma using graphs only (Figure 1 specifically). They can make a better comparison by running the same interrupted time series model for asthma or a non-tobacco-related disease, which I recommend, and see if the TCL has had any impacts on the control.
Response 2: Method
- We agree that an interrupted-time-series-year-fixed-effects model may be an option, and adding region fixed effects to the model may be beneficial due to the potential variation in effects across regions. However, we hypothesized that the interrupted time series model we implemented is also suitable for this study, since it is usually not affected by typical confounding variables, such as population age distribution or socioeconomic status, as they remain fairly constant and change relatively slowly over time. Regarding differences between regions, potential confounders changing across regions were controlled for by including variables representing them in the regression model.
- Following the recommendation, we made another comparison by running the same interrupted time series model for a non-tobacco-related disease, viz., rheumatic heart disease, which is just as well controlled at the outpatient level as asthma. The graph was added to Figure 1. In addition, we have added the results of the same interrupted time series model for asthma and rheumatic heart disease to the figures. The analysis revealed no statistically significant effect of TCL on the control (lines 200 – 202).
Point 3: Results
The findings about the correlation between hospital admissions for pneumonia and different components of the TCL provide nice support to the conclusion. However, I recommend that the authors separate the main findings and the mechanism findings since the models used for the estimation are different, and also the channels of the effect should be highlighted more in the paper.
Response 3: Results
Two more paragraphs have been added to the Results section to clarify our findings and how they were obtained: (lines 217-220; 246-249).
